# Potential Role of Birds in Japanese Encephalitis Virus Zoonotic Transmission and Genotype Shift

**DOI:** 10.3390/v13030357

**Published:** 2021-02-24

**Authors:** Muddassar Hameed, Abdul Wahaab, Mohsin Nawaz, Sawar Khan, Jawad Nazir, Ke Liu, Jianchao Wei, Zhiyong Ma

**Affiliations:** 1Shanghai Veterinary Research Institute, Chinese Academy of Agricultural Science, Shanghai 200241, China; abdul.wahaab@uaf.edu.pk (A.W.); mohsin4846@yahoo.com (M.N.); drsawarkhan@gmail.com (S.K.); liuke@shvri.ac.cn (K.L.); 2Vaccinologist/Head Virology, Tréidlia Biovet Pty Ltd. Units, Seven Hills, NSW 2147, Australia; jawad.nazir@uvas.edu.pk

**Keywords:** Japanese encephalitis virus, birds, genotype shift, JEV genotype I, JEV genotype III

## Abstract

Japanese encephalitis (JE) is a vaccine-preventable disease caused by the Japanese encephalitis virus (JEV), which is primarily prevalent in Asia. JEV is a Flavivirus, classified into a single serotype with five genetically distinct genotypes (I, II, III, IV, and V). JEV genotype III (GIII) had been the most dominant strain and caused numerous outbreaks in the JEV endemic countries until 1990. However, recent data shows the emergence of JEV genotype I (GI) as a dominant genotype and it is gradually displacing GIII. The exact mechanism of this genotype displacement is still unclear. The virus can replicate in mosquito vectors and vertebrate hosts to maintain its zoonotic life cycle; pigs and aquatic wading birds act as an amplifying/reservoir hosts, and the humans and equines are dead-end hosts. The important role of pigs as an amplifying host for the JEV is well known. However, the influence of other domestic animals, especially birds, that live in high abundance and close proximity to the human is not well studied. Here, we strive to briefly highlight the role of birds in the JEV zoonotic transmission, discovery of birds as a natural reservoirs and amplifying host for JEV, species of birds susceptible to the JEV infection, and the proposed effect of JEV on the poultry industry in the future, a perspective that has been neglected for a long time. We also discuss the recent in vitro and in vivo studies that show that the newly emerged GI viruses replicated more efficiently in bird-derived cells and ducklings/chicks than GIII, and an important role of birds in the JEV genotype shift from GIII to GI.

## 1. Introduction

JEV causes neurological disease, which is one of the leading types of viral encephalitis in the world [1]. According to the recent estimated global incidence of Japanese encephalitis, approximately 67,900 JE cases occur per year in the 24 JE endemic countries, for an incidence of 1.8 per 100,000 [1]. Approximately 33,900 (50%) of these cases occur in China and 55,000 (81%) occur in areas with well-established or developing JE vaccination programs, while approximately 12,900 (19%) occur in areas with minimal or no JE vaccination programs. Approximately 51,000 (75%) of these cases occur in children aged 0–14 years, which gives an estimated overall annual incidence of 5.4 per 100,000 in this age group [1].

The majority of human infections are asymptomatic, and many symptomatic cases results in meningitis, encephalitis or flaccid paralysis, and are fatal or cause devastating long-term neurological sequelae. JEV epidemics were originally reported from Japan in the nineteenth century, and the virus was isolated for the first time in 1935 from an infected human brain sample in Tokyo [2]. JEV infections occur across a large range of Asian countries with an outbreaks occurring in Japan, China, Taiwan, Korea, the Philippines, and India [3]. JEV cases occur in Nepal, India, Papua Guinea, Pakistan, and Australia, suggesting that this virus is going to expand its geographic range in the future [4,5]. In 2017, a whole JEV genome was identified by unbiased RNA sequencing in a patient coinfected with yellow fever in Cunene province, Angola, raising the possibility that the geographic range of JEV might be greater than previously thought [6]. This shows that JE might be going to become a public health problem of intercontinental concern [7].

JEV has a positive sense RNA genome belonging to Flavivirus genus within *Flaviviridae* family, harboring three structural and seven non-structural proteins. JEV life cycle contains both invertebrates (mosquitoes) as well as vertebrates (wild birds and pigs). On the basis of phylogenetic investigations, JEV is classified into a single serotype with five genetically distinct genotypes (GI, GII, GIII, GIV, and GV). GIII had been the most prevalent strain, with a number of epidemics in the past. However, recent studies report the emergence of GI strain as a leading JEV genotype [8]. For a number of years, JEV isolates from Japan, Republic of Korea, and China were sorted out under JEV GI, although these regions were having GIII endemic history [8,9]. In addition, the more divergent genotype V strains (amino acid divergence from 8.4% to 10.0% compared to genotypes I–IV) have been detected in Malaysia [10], Korea [11] and China [12], and may be covered poorly by the existing JEV genotype III-based vaccines [13,14,15]. There is also concern that JEV could spread to the Americas and Europe, much like the West Nile virus (WNV) did, as North American field-collected *Culex* mosquitoes and experimentally exposed *Culex* mosquitoes from Europe were found susceptible to the JEV infection [16,17,18]. Furthermore, several avian species in North America are susceptible to JEV and can possibly serve as an amplification hosts [19]. The spread of arboviruses such as WNV and JEV can occur by wind-blown mosquitoes, migrating viremic birds or anthropogenic activities [7,20].

Pigs’ role as an amplification host for JEV has been well demonstrated in previous studies [21]. Seroprevalence surveys report that 3.1% to 74% anti-JEV antibodies were present from 1966 to 2016 in swine herds in most of the Asian countries [21]. Pigs can develop a high level of viremia that is maintained for two to four days and attracts the JEV vectors, not only for the virus maintenance/amplification, but also for the pig–mosquito–pig or pig–mosquito–human transmission [22,23,24]. Furthermore, pig farms are mostly located close to human dwellings, especially those with backyard farming, which is common in Asia; this proximity facilitates the human infection. A total of four types of JEV vaccines are available in different countries, including a live attenuated vaccine, a live recombinant (chimeric) vaccine, inactivated mouse brain-derived vaccines, and inactivated Vero cell culture-derived vaccines [25,26,27,28,29]. Both JEV inactivated and live-attenuated vaccines are used in pigs to prevent JEV infection in different Asian countries [25,30,31].

Various other species of domestic animals, such as cattle, dogs, cats and wild boars, can be sub-clinically infected, as evidenced by reports of seropositivity to the JEV in surveys [30,32,33,34,35]. Previously, a number of seroepidemiological surveys of wild boars antibodies against JEV have reported that a higher percentage of wild boars sera contained JEV antibodies [36]. On Iriomote Island, 44.4% (52 out of 117) of wild boars’ serum samples tested positive for the presence of JEV antibodies [33,34,37,38,39]; in Hiroshima, 68% (17 out of 25); in Wakayama, 83.3% (30 out of 36); in Ishikawa prefecture, 86.5% (32 out of 37); in Singapore, 15.16% (10 out of 66). Wild boars are closely related to domestic pigs and are not inoculated with the JEV vaccines; they are thought to play a similar role as a reservoir in the infection cycle of JEV. Recently, JEV GI strains were isolated and characterized from a nonsuppurative encephalomyelitis case of 141-day-old calf and 114-month-old cow from Japan, suggesting that JEV GI infection can lead to the development of neurological disorders in cows [40,41]. These animals do not act as a reservoir/amplification hosts for JEV but can play a potential role in future and continuous monitoring is required.

In addition to pigs as a main JEV reservoir, some studies suggested that other epidemiological systems may exist. A recent autochthonous human JEV case occurred in Seoul, South Korea, even though no pigs are reared in the city [42]. In 2013, Teng et al. isolated JEV strains from mosquitoes, humans, and pigs, and revealed that the same strain was detected in mosquitoes and humans, but not in pigs [43]. Two other studies from Singapore Islands showed that the JEV has continued to circulate decades after the abolition of pig farming [39,44]. Thus, there is a need to improve our knowledge of the existence of secondary JEV reservoir hosts that could explain JEV transmission in areas with no or low pig density. Additionally, the JEV genotype shift from GIII to GI has reported in some countries where pig-farming is not common, such as Malaysia [45], India [46], and Korea [42]. Human infection does not contribute to the JEV transmission and the human vaccine does not reduce transmission of JEV in the reservoir community, no herd immunity is generated, and vaccination has to be continued indefinitely. Birds’ role as an amplification/reservoir host has been poorly investigated. Therefore, in this review, we have highlighted the role of birds in the JEV transmission, discussed the role of birds in JEV genotype shift from GI to GIII, and the proposed effect of JEV on poultry industry in future perspective. Thus, we can contain the JEV spread while taking possible countermeasures that can blunt their impact on public health as well as for veterinary concerns.

## 2. JEV Zoonotic Transmission

Mosquito-borne zoonosis includes JEV life cycle involved both invertebrates (mosquitoes) as well as vertebrates (wild birds and pigs). The JEV is transmitted by several *culicine*, *aedes*, *anopheles*, and *armigeres* mosquito species. *Culex* (*Cx*.) species mostly involved in the transmission cycle of JEV. In initial investigations from Japan, relative abundance of each mosquito species caught in baited traps and their JEV infection status were compared when implicating vectors in transmission [47]. It was noticed that *Culex tritaeniorhyncus* is the primary vector for JEV transmission, which was strengthened later by laboratory experiments indicating this mosquito’s competence for JEV replication and transmission [47,48]. However, recent studies shows that other *Cx.* species are also competent vector for JEV [17,18,49].

In addition to mosquitoes as a vector, pigs and ardeid birds act as an amplifying/reservoir host [50]. Theoretical models of vector-borne pathogen transmission demonstrate that the pathogen transmission rate particularly depends upon the proportion of vector blood meals taken from the competent hosts versus dead-end hosts [51]. Usually, JEV is transmitted from infected pig/bird to non-infected pig/bird/human by mosquitoes, but recently it has been reported that this could be independent of the vector in pigs [52]. Pigs serve as amplifying hosts because they can develop sufficient viral titers to support further infection of mosquitoes [21].

Previous studies reports the dominance of pigs as an amplifying hosts, but recently this concept has been challenged, as some countries, such as Bangladesh, which has very small pig population, also have an appreciable burden of Japanese encephalitis in humans [53]. This reveals the presence of some other potential hosts that are amplifying JEV. Recently, we have detected JEV in mosquitoes collected from different animal farms during arboviral surveillance located in Xinjiang, China, which also has a small pig population [54]. Although the role of birds as a reservoir hosts for JEV is admitted the role of birds as potential hosts has been poorly investigated in the past.

Unlike in pigs, no onward transmission occurs from humans because JEV induced viremia is insufficient to be infectious to the mosquito vector, making humans dead-end host for JEV; therefore, mosquitoes cannot get from an infected person [55].

## 3. Discovery of Birds as the Natural Reservoirs and Amplifying Host for JEV

The role of birds as a reservoir hosts for JEV has been admitted since 1958 [48,56]; however, the role of birds as potential amplifying hosts has been little investigated so far. Several surveys conducted in different continents suggest the involvement of domestic birds in arboviruses dispersion; especially ducks, as involved in WNV epidemiological cycle, either as an amplifying host or as a reservoir [57,58,59], which is the most closely related to JEV among flaviviruses and shares ecological resemblance. Both of these viruses maintain their enzootic transmission cycle with several bird families as a natural reservoir and mosquitoes of *Cx.* species as a main vector [20,60]. With regard to JEV, a number of studies have been conducted in birds to determine the seroprevalence for JEV. For example, Saito et al. suggested that wild ducks can play an important role as a JEV reservoir hosts [61]. Saito and his colleagues captured 92 wild ducks; 50 Anas platyrhynchos (undetermined), 16 Anas acuta (winter visitors), 6 Anas penelope (winter visitors), and 20 Anas poecilorhyncha (migratory breeders) in autumn of 2005 and 2006, in the central part of Hokkaido, a low JEV activity area. They performed seroepidemiologic analysis of JEV and tested 5.4% and 85.9% positive for JEV-specific antibodies with 90% (FRNT 90) and 50% focus reduction neutralization tests (FRNT 50), respectively [61]. In addition, Yang et al., reported that out of 1316 serum samples tested, 84.7% to 88.5% sero-prevalence for JEV in wild birds including ducks (Anas penelope, Anas formosa, Anas crecca, Anas acuta, Anas poecilorhyncha, Anas platyrhynchos), petrels (Oceanodroma castro), mandarin ducks (Aix galericulata), and Eurasian coots (Fulica atra) during 2009 from Korea [62]. In Bali (Indonesia), there were 20.6% of ducks and 36.7% of chickens were tested positive [63]. Furthermore, a study from Malaysia found 28.9% of the tested domestic birds positive for the JEV antibodies [64]. A recent study from Cambodia reports 29% (180/620) of the domestic birds positive for flavivirus antibodies with an age-depended increase of the seroprevalence (OR = 1.04) and a higher prevalence in ducks compared to chicken (OR = 3.01) [65]. Within the flavivirus positive birds, they found 43% (28/65) with nAb against JEV [65].

Along with this, a number of recent experimental studies shows that domestic birds can be infected with JEV [48,66,67,68] and might even act as a JEV reservoirs [69,70]. A recent study from Korea reports that the distribution and the density of migratory birds are correlated with JE cases in cities and they might be highly potential hosts contributing to transmit JEV in metropolitan areas [42]. Because of the bird’s close association to humans and varying levels of seroprevalence observed in birds, their role in epidemiological cycle as secondary reservoirs for JEV may be of importance.

## 4. Species of Birds Susceptible to JEV Infection

The inter-continental spread of JEV and other arboviruses to non-endemic areas is a continual impeding threat [71]. The circulation of JEV in the Southeast Asia is well-documented, and the important role of pigs as an amplification hosts for the virus has been well known from long time. However, presently, pigs might be playing a less important role as an amplifying hosts as compare to past because of the JEV vaccination and vector control at farms. The influence of domestic and wild birds that live in high abundance and close proximity to human and animals is not well studied. Similar to the unanticipated spread of WNV in America, the geographic range of JEV has expanded within the past decade.

Species within the avian family *Ardeidae,* e.g., herons, bitterns, and ergets, were initially targeted for JEV research because they are seasonally abundant and available, and are relatively easy to sample. Subsequent high seroprevalence and JEV isolation from ardeids in Japan and India gives an idea that they may have an important role in JEV dispersion [56,72]. Characterization of avian host species response to JEV infection and seroprevalence is an important fact to elucidate the birds’ role as a reservoir host. Differences in JEV viremia profiles among variety of avian species were observed in a number of studies, which we discuss below.

Nemeth et al. observed variation in interspecies responses among the North American birds during 2011–2012. They used 16 species of birds from eight taxonomic orders as shown in Table 1 [19] to JEV infection. Nemeth’s team noticed that the majority of individuals of all species inoculated with JEV genotype I or III had the highest average peak viremia titers except for fish crows, ring-necked pheasants, American crows, American white pelicans, and double-crested cormorants; no individuals of these species had a detectable viral load. Conversely, the majority of the birds, both viremic (72 of 74; 97.3%) and non-viremic (31 of 37; 83.8%), were seroconverted by 14 days post-inoculation [19].

In 2014, Cleton et al. investigated the magnitude of virmia in 2-day-old chicks and ducklings after JEV infection [68]. In their study, 2-day-old chicks and ducklings were represented with peak viremia at 3 days post infection as presented in (Table 1) [68]. In addition, infection was associated with reduced weight gain in both species, and ducklings infected at 10 days of age or less showed overt clinical signs of disease. Furthermore, the mean peak viremia in birds of both species decreased as the age at infection increased from 2 to 42 days, indicating the importance of age on magnitude of viremia in birds from both species, and suggesting that young poultry birds may be amplifying hosts of importance in disease-endemic regions.

Following Cleton et al.’s study, we have examined the pathogenicity of JEV strains (SD12, SH1, SH2, SH7, SH15, SH19, and N28) in the Shaoxing ducklings at day 2 post hatching as shown in Table 2 [66]. After subcutaneous inoculation with 10,000 plaque-forming units of JEV per bird, all ducklings were monitored for 7 days and weighed daily from 0 dpi to the end (7 dpi) of experiment to calculate the average daily weight gain (ADWG). A blood samples were taken from jugular vein at 2 dpi for the detection of viremia by 50% tissue culture infectious dose (TCID50) assay.

Some JEV-inoculated birds showed mild and non-characteristic clinical signs starting from 2 dpi. The ADWG of all JEV strain-inoculated ducklings was significantly lower as compared to that of mock-inoculated ducklings during the 7-day experiment, with reductions of 2.1–4.5 g suggesting stunted growth in the JEV-inoculated ducklings [66]. No death was observed in ducklings challenged with SD12, SH1, SH7, or SH15. In contrast to this, significant mortality was observed in ducklings inoculated with JEV strains N28 (31.7%) and SH19 (12.7%) [66]. Furthermore, the proportions of viremic ducklings and the viremia titers differed among strains, with the highest proportion (69.2%) of viremic individuals and the highest viremia titer (10^3.4 ± 1.3^) in ducklings inoculated with SD12 strain (Table 2). These results suggest that the response and susceptibility of ducklings to JEV infection differed among JEV strains [66]. Along with this, a previous study also reported JEV-induced death in experimentally-infected wild birds of several species [75].

In another study, Karna and his team infected 5 to 6 days of age Indian runner ducks (Anas platyrhynchos domesticus) with ~10^6^ PFU of JEV using 6 different strains from JEV genotype I and III (Table 2). The mean peak viremia titer developed in inoculated ducklings at 2–3 dpi against six strains of JEV GI and GIII (mean ± 1 SE, log10PFU/mL) were 3.9 ± 0.2 and 3.5 ± 0.3, respectively [73]. However, none of the ducklings presented with signs of disease or distress [73]. Notably, in another experiment, we have inoculated Shaoxing ducklings at day 2 post-hatching to compare replication efficiencies between JEV GI and GIII strains [67]. All injected ducklings had developed viremia with similar viremic rates between GI and GIII, while the viremic duration of GI-inoculated ducklings was notably, but not significantly, longer than GIII-inoculated ducklings (Table 2) [67]. These data further support the claim that JEV infection leads to the development of viremia in birds.

Results from all the above-mentioned studies revealed that relatively small JEV doses injected subcutaneously into birds resulted in infection, implying that relatively low viral quantities injected into birds by mosquitoes could result in infection. However, this has an important caveat that all the above-mentioned investigations lack a natural route of infection in birds by mosquitoes, because the difference in inoculation route could lead to difference in pathogenicity of the same JEV strains. The components of mosquito saliva play roles in modulating host immune responses and in facilitating the replication and transmission of flaviviruses [76,77,78]. Furthermore, the amount of virus present in host blood after a bite by an infectious mosquito is also an important parameter in determining the extent to which a host may contribute to transmission [79].

Recently, we have noticed 30% mortality in newly hatched Shaoxing ducklings when bitten by infected mosquitoes under lab conditions [80], which was not seen in the domestic ducklings subcutaneously inoculated with the same JEV strain used in our previous study [66]. The infected ducklings died suddenly, with neurological signs of opisthotonos (a condition of spasm of the back muscles causing the head and limbs to bend backward and the trunk to arch forward) between 2 and 3 dpi. The highest RNAemia were observed in the affected ducklings at 2 and 3 days infected mosquito bite [80]. However, the remaining ducklings exposed to JEV-infected mosquitoes showed no noticeable clinical signs. This apparent difference in the pathogenicity of the same JEV strain may be attributable to the difference in the inoculation route between the two experimental challenges. These observations indicated that JEV infection via mosquito bite causes mortality associated with viral encephalitis in newly hatched ducklings, thus demonstrating the potential pathogenicity of JEV in domestic ducklings under natural conditions. The viremia found specifically in the young birds were high enough that can resulted in 50–100% transmission in *Culex* mosquitoes [81,82], which are a known competent vector for the JEV [49].

Considering these estimates for the vector competence, these studies suggest that efficient transmission of JEV to mosquitoes likely occurs from young chicks, ducklings and other bird species, may play an epidemiologically significant role in JEV transmission. Birds’ potential for JEV to spread to non-endemic areas and potential impact of particular farming systems, including duck farming needed further investigation. Along with this, field studies to explore the force of infection in these hosts during JEV transmission events are necessary to further validate their role in the JEV transmission dynamics.

## 5. Pathogenicity of JEV in Domestic Birds

JEV-induced mortality were reported in experimentally inoculated wild birds of several species [75]. Previous experiment data show that inoculation of domestic ducklings with different JEV strains resulted in overt clinical signs, stunted growth and variable viremia in all JEV-inoculated ducklings [66,68], suggesting the potential pathogenicity of JEV in newly hatched domestic ducklings. Although no JEV-related outbreaks have yet been reported in domestic ducklings, recent findings suggest that the responses and susceptibilities of ducklings to the JEV infection is age-dependent and differ among JEV strains. Among the seven JEV strains used by Xiao and his colleagues, only one strain showed high virulence in ducklings, and a proportion of ducklings failed to develop detectable viremia after JEV inoculation, suggesting that most of the JEV strains circulating in natural hosts might have low or nonexistent pathogenicity in domestic ducklings. In addition, natural JEV infection via JEV-infected *Cx. pipiens* mosquito bites in newly hatched domestic ducklings caused 30% mortality, which was associated with viral encephalitis [80]. These observations demonstrated the potential pathogenicity of JEV in domestic ducklings under natural conditions. Presently, it is also possible that any JEV outbreak in the ducklings might be ignored or misdiagnosed because of the mild and non-characteristic clinical signs and the relatively low mortality. In future, surveillance of ducklings dying in the mosquito season in JEV-endemic areas is needed to elucidate the potent pathogenicity of JEV in poultry birds and possible prophylactic strategies we should take to avoid its outbreaks.

## 6. In Vitro and In Vivo Studies Shows the Potential Role of Birds in JEV Genotype Shift

GIII was an endemic strain in Asia but, recently, GI has displaced GIII as the most frequently isolated virus genotype. The exact mechanism that led to this genotype shift is still not clear. In past, there have been a number of reports about the isolation of JEV GI from human, mosquitoes, and pig samples. A recent study reported the JEV outbreak among human caused by GI in Ningxia in the Northern China [83]. This study confirmed the JEV GI outbreak by isolating G I from laboratory (human) and field data (mosquitoes).

Birds play an important role in the maintenance and transmission of many arboviruses including JEV. As we have discussed above, avian species can develop viremia against JEV after either natural exposure or challenged in lab [7,19,20]. These reservoir hosts may have some important role in the JEV genotype shift that can be explored. In a previous study, we observed that JEV GI strains replicated more efficiently than GIII strains particularly in birds derived cells and in the young ducklings [66,67]. This shows that JEV GI has an advantage in replication efficiency and host adaptation in birds which can lead to JEV genotype shift. However, the mechanism behind this adaptation in birds required further investigation.

In a recent study from Taiwan, Fan Y.-C. and his coworkers compared JEV GI and GIII infectivity in one day old chickens and two day old ducklings [74]. Fan Y.-C. et al. inoculated 10^4^ FFU of GI and GIII viruses in one day old chicks and two day old ducklings. They observed 100% (8/8) and 75% (6/8) viremia in GI infected chicks and ducklings (Table 2); whereas, 37.5% (3/8) and 12.5% (1/8) of GIII inoculated chicks and ducklings had developed viremia [74]. Their data demonstrated that JEV GI infected birds showed a significantly higher viral titer (0.60–1.73-log) as well as earlier and long lasting viremia than birds inoculated with GIII strains as shown in Table 2.

They had performed a series of experiments by using chimeric viruses (GIII/GI UTR, GIII/GI C-E, GIII/GI NS1-5, GIII/GI NS1-3, GIII/GI NS4-5) and demonstrated that the higher GI virus infectivity determinants are present in the NS1-3 genes of the JEV genome. They further verify the specific substitutions of GI NS1-3 protein by introduction of a single virus specific and highly consensus substitution in rGIII/GI NS1-3 chimeric viruses. Their experimental data concluded that the GI residues NS2B-V99L and NS3-A78S, NS3-E177D were involved in the replication enhancement of GI virus in vitro (DF-1) and in vivo (1-day-old chicks and 10-week-old SPF pigs) (Table 3).

Recently, we had conducted a deep investigation to identify the viral determinants of differing multiplication capability between GI and GIII viruses in birds. We examined the difference in Interferon –I (IFN_I) stimulation between GI and GIII by using duck embryo fibroblasts (DEF) and domestic ducklings as an in vitro and in vivo avian models, respectively [84]. The DEF, mouse endothelial cell line (bEnd.3), and swine testicular cells (ST) were infected with GI (SH7 and SD12) and GIII (SH15 and SH19) strains to analyze the induction of IFN- α and β expression. INF- α and β production was significantly lower in DEF cells infected with GI strains as compare to GIII viruses; whereas, no significant difference was seen in the IFN- α and β expression in the ST and bEnd.3 cells infected with GI and GIII strains. This species-specific IFN expression by GI and GIII viruses was also confirmed by infection duck kidney cells (DEK), porcine iliac endothelium cell line (PIEC), and mouse embryonic fibroblast cell line (MEF). Similarly, GI strains decreased the IFN expression in DEK cells, whereas there was no statistical difference seen in PIEC and MEF cells after GI and GIII infections. GI strains capability to stimulate low levels of IFN- α and β expression was further confirmed in the domestic ducklings. GI viruses produced viral titers 0.5–1 log higher than GIII in DEF cells due to IFN-I mediated antiviral response [85,86].

We had used a series of chimeric recombinant viruses with the exchange of structural and non-structural proteins between the GI and GIII strains, and identified the NS5 gene as the viral determinant of the differences in IFN- α and β expression and replication efficiency between the JEV strains in ducklings. GI and GIII virus genetic analysis reveals that the NS5 gene contained a total of 11 amino acid variations. We performed a series of chimeric substitution mutations and identified that NS5-V372A and NS5-H386Y variations co-contribute to the differences in IFN- α and β expression induction and replication efficiency between the JEV GI and GIII strains in DEF cells and ducklings (Table 3). Then, we investigated the role of NS5-372 and NS5-386 substitutions in GI and GIII strains conformation.

The substitution analysis revealed that NS5-372 makes two hydrogen bonds and NS5-386 form one hydrogen bond in GI strains with their neighboring residues, respectively. On the other hand, GIII NS5-372 substitution makes three hydrogen bonds and NS5-386 leads to the formation of two hydrogen bonds. This bonding difference in GI and GIII strains potentially result in the variation of flexibility of the NLS (nucleus locating signals) region, ultimately leading to changes in interactions with the host cell proteins and IFN-I production and viral replication. The differences in replication efficiencies, IFN-α and β production among GI and GIII strains, were detected only in duck-derived DEF and ducklings, but not in pig-derived ST cells and mouse-derived bEND.3 cells. There are two possible reasons for this difference; first, the changes in the antiviral immune response between birds and mammals, and, second, the differences in adaptability of GI and GIII viruses to the IFN-I mediated antiviral response of birds and mammals. The NS5-V372A and NS5-H386Y variations allowed GI viruses to adapt the IFN-I mediated antiviral immune response of birds, but not mammals, thereby leading to the replication advantages of GI strains over GIII in birds. Although this experiment is lacking the in vivo data after infection in pigs with these recombinant viruses; however, the above-described results explain the host specific differences in the IFN-I induction among GI and GIII strains leads to the replication and host advantages of GI over GIII viruses in birds, that might be a cause of JEV genotype shift.

Overall, our current knowledge about the role of birds in JEV genotype shift suggest that it is important to continually monitor JEV GI virus evolution and role of birds in local transmission of JEV GI viruses.

## 7. Role of JEV Vaccines

Although the introduction of JEV inactivated and live attenuated vaccines had dramatically reduced the JE cases, JEV still remains a leading cause of viral encephalitis globally. Recent detection of more divergent JEV genotype V (8.4% to 10.0% amino acid divergence compared to genotype I-IV) from China [12] and Korea [11] is threatening because it may be covered poorly by the presently used JEV GIII strain-based vaccines [13,14,15].

Mostly, JEV vaccine applied at all pig farms as a regular vaccine campaign [25,30,31]. Therefore, quantifying the relative contributions of pigs and domesticated birds to JEV transmission is required for understanding the recent JEV ecology in regions where the pigs mostly vaccinated or pig population density is relatively low compared to the birds’ population density. As we have discussed above, ducks, chickens, pigeons, and other birds produced viremia following JEV infection, which demonstrates their role as a JEV amplifying/reservoir hosts [19,48,66,67,68,73,80,81,87]. JEV infection produced viremia in these birds, which is sufficient to infect mosquitoes, but their contribution to the JEV transmission remains to be quantified.

We are proposing a hypothesis that should be evaluated as shown in Figure 1; (i) presently, pigs may be contributing less due to wide application of JEV vaccines at pig farms than birds to JEV transmission and genotype shift, and (ii) JEV GI shows higher replication efficiency than GIII in duck-derived cells and in ducklings/chicks. Due to lack of JEV vaccination, birds can play an important role in the JEV transmission and genotype shift from GIII to GI. There are, however, currently insufficient data to fully assess this hypothesis, and further study is required.

## 8. Concluding Remarks

JEV has become a significant global pathogen that is causing major public health problems in Asia. Overall, in the past, the role of pigs in the JEV epidemiology has been investigated deeply as these are well-known amplification hosts for this virus. However, the contribution of birds to the JEV transmission remains ignored. Previous studies reported that avian species can develop viremia after either natural exposure or being challenged in the lab [7,19,20] and can develop clinical signs, which are ubiquitous, often sharing urban and suburban habitats with the humans and mosquitoes. These amplifying/reservoir hosts may have an important role in the expansion of JEV affected areas which needs further investigation.

The dominant genotype of JEV has changed from III to I around 1990, and the mechanism behind this genotype shift is still unknown. Recent studies demonstrated that JEV GI had superior replication activity in birds’ derived cells as well as in young ducklings and chicks. The proposed molecular mechanism is the variation of NS2B-V99L, NS3-A78S, NS3-E177D and NS5-V372A and NS5-H386Y genes among GI and GIII viruses. These substitutions lead to the replication advantages of GI strains over GIII strains. This also emphasizes the need for further and intensified monitoring of JEV GI evolution in birds. In addition, surveillance of JEV in backyard domestic poultry and migratory birds that serve as potential amplification hosts is required, which will help focusing preventive measures, such as vaccination and vector control, in the future.

## Figures and Tables

**Figure 1 viruses-13-00357-f001:**
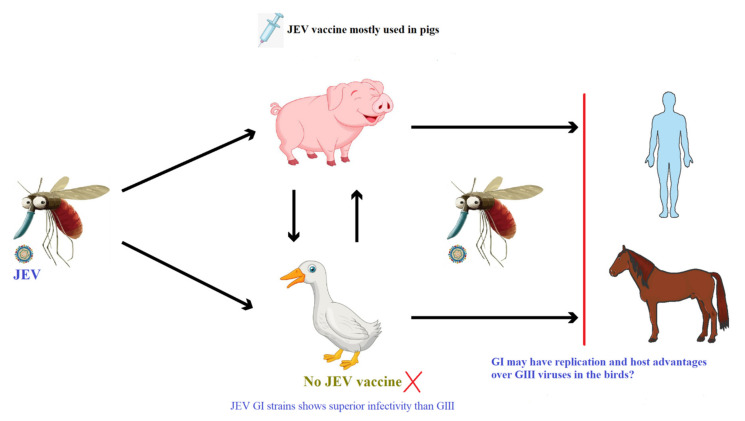
Overview of JEV transmission cycle and the expected role of the JEV vaccine application.

**Table 1 viruses-13-00357-t001:** Summary of viral titers of birds experimentally inoculated with Japanese encephalitis virus.

Birds	Level (log10 Plaque-Forming Units/mL)	Reference
Fish crow	<10^1.7^	[19]
Ring-necked pheasant	<10^1.7^	[19]
Mallard	10^2.0–3.3^	[19]
House sparrow	10^1.7–3.7^	[19]
Red-winged blackbird	10^2.3–4.0^	[19]
Rock pigeon	10^2.7–4.3^	[19]
European starling	10^2.5–3.6^	[19]
House finch	10^3.8–4.9^	[19]
Common grackle	10^3.3–4.4^	[19]
Ring-billed gull	10^3.5–5.4^	[19]
Cattle egret	10^2.0–3.1^	[19]
American crow	<10^1.7^	[19]
American white pelican	<10^0.7^	[19]
Double-crested cormorant	<10^0.7^	[19]
Chicken	10^1.7^	[19]
Great egret	10^3.4–4.2^	[19]
Chicks $	10^4.7^	[68]
Ducklings ¥	10^6.3^	[68]

$ values of highest viral titer after 3 dpi in 2 day old chicks. ¥ values of highest viral titer after 3 dpi in 2 day old ducklings.

**Table 2 viruses-13-00357-t002:** Viral titers of birds inoculated with the Japanese encephalitis virus GI and GIII strains.

Genotype	Strains	Birds	Titer (log10 Plaque-Forming Units/mL)	Reference
	SD12	Ducklings	10^3.2 ± 0.7 ±^	[66]
	SH2	Ducklings	10^2.8 ± 0.9 ±^	[66]
	SH7	Ducklings	10^2.5 ± 0.6 ±^	[66]
	KE-093-83	Ducklings	10 ^(4.1 ± 0.2)^	[73]
	MAR864	Ducklings	10 ^(3.3 ± 0.2)^	[73]
	JE-91	Ducklings	10 ^(4.2 ± 0.1)^	[73]
**GI**	SH2	Ducklings	0.80 ἀ	[67]
	SH7	Ducklings	2.25 ἀ	[67]
	SD12	Ducklings	0.2 ἀ	[67]
	YL2009-4	Chicken	6.0 €	[74]
	TC2009-1	Chicken	6.25 €	[74]
	YL2009-4	Ducklings	4.30 €	[74]
	TC2009-1	Ducklings	4.30 €	[74]
	SH1	Duckling	10^2.2^ ^± 0.7^	[66]
	SH15	Ducklings	10^2.0^ ^± 0.4^	[66]
	SH19	Ducklings	10^2.9^ ^± 0.8^	[66]
	N28	Ducklings	10^2.5^ ^± 1.1^	[66]
	CH392	Ducklings	10 ^(3.0 ± 0.8)^	[73]
	JKT27-087	Ducklings	10 ^(4.2 ± 0.3)^	[73]
**GIII**	Sagiyama	Ducklings	10 ^(3.3 ± 0.6)^	[73]
	N28	Ducklings	0 ἀ	[67]
	SH1	Ducklings	0 ἀ	[67]
	SH15	Ducklings	0.2 ἀ	[67]
	SH19	Ducklings	0 ἀ	[67]
	CH1392	Chicken	4.75 €	[74]
	T1P1	Chicken	5.10 €	[74]
	CH1392	Ducklings	4.10 €	[74]
	T1P1	Ducklings	3.40 €	[74]

^±,^ Viremia titer was tested by TCID_50_ assay (TCID_50_/0.1 mL) at 2 dpi. €, FFU titer at 2 days post infection. ἀ TCID50 viral titer measured at 4 days post infection.

**Table 3 viruses-13-00357-t003:** The amino acid variations conserved between JEV GI and GIII strains and some key substitutions at NS2B/NS3/NS5 that might be helping GI viruses to adapt reservoir/amplification host and have effect on JEV transmission.

Protein	GI Residue/Position/GIII Residue											NS2B/NS3/NS5 Substitutions ^$^
C	R70K ^a^	K100R ^a^	S110G ^a^	I120V ^a^	T122I ^a^							
PrM	A57T ^a^	V58M ^a^	A140V(I) ^b^	S149N ^a^								
E	M129T ^a^	S222A ^a^	T327S ^a^	S366A ^a^								
NS1	Q51K ^a^	S70A ^a^	R147H ^a^	N(D)175S ^b^	L206Y(F) ^b^	R251K ^a^	I298V ^a^					
NS2A	I6V ^a^	A97T ^a^	T149S ^a^	A151T ^a^	R187K ^a^							
NS2B	D55E ^a^	E65D ^a^	L99V ^a^									V99L
NS3	L14S ^a^	S78A ^a^	P105A ^a^	I(V)123V ^b^	D177E ^a^	S182N(T) ^b^	K185R ^a^	D354E ^a^				A78S, E177E
NS4A	V110I ^a^											
NS4B	R(K)20K ^b^	P(S)24S(P) ^b^	S73N ^a^	V118A ^a^								
NS5	D22E ^a^	K101R ^a^	R280K ^a^	R(K)287K ^b^	V372A ^a^	H386Y(H) ^b^	G429D ^a^	L432R ^a^	N438D ^a^	G588E ^a^	I878V ^a^	V372A, H386Y

^a^, Amino acid variation with conservation rate of 90–100% in GI and GIII strains. ^b^, Amino acid variation with conservation rate of 50–89% in GI and GIII strains. $, The contribution of NS2B/NS3/NS5 substitutions on replication advantage of GI virus in amplifying hosts.

## Data Availability

Not applicable.

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
