# Peer review of "Potential Role of Birds in Japanese Encephalitis Virus Zoonotic Transmission and Genotype Shift"

_viruses, 2021, doi:10.3390/v13030357_

Round 1
Reviewer 1 Report
In this review manuscript the authors focus to the importance of birds as amplifying/reservoir host of JEV and also as key animal in JEV genotype shift from GIII to GI. The review also introduces the historical researches for discovery of birds as the natural amplifying/reservoir host of JEV and for evaluation of sensitivity of birds to JEV infection. The authors also refer to the possibility that birds may be committed mainly to GIII to GI shift in JE-endemic areas.
1) First, the reference numbers in the manuscript and Table 1 are inconsistent with those of the references list. The authors should re-check all of the citations in the manuscript.
2) Only summarized results and discussion of the paper from the authors group (Li C et al #65) are described in paragraphs 9-11 of section 5 (line 335-379). In the paper, the authors did not perform in vivo JEV infection analysis in pigs (only used pig-derived cultured cells). In vitro growth ability of JEV is necessarily reflect its growth ability in vivo. Fan et al. (#64) showed that GI JEV grew higher than GIII JEV in not only chickens also pigs. Therefore, it is hard to conclude that birds play main role in JEV transmission and genotype shift in most JE-endemic areas.
3) Table 1 should be reconstructed (made smaller). The column “age” is indispensable? Data of birds infected with each JEV genotype strains should be combined into one row (group). The author should be confirmed the references.
4) Figure 1 need to be reconstructed. Is there critical evidence that major amplifying host of JEV is now birds not pigs in most JE-endemic areas? In my understand, pigs as well as birds are reservoir and amplifying hosts of JEV and other mammals (cattle, wild boar, bat etc.) also have potential to reservoir/amplifier of JEV. “JEV vaccine mostly used in pigs” is correct? In China and other areas most sentinel pigs are vaccinated? Many papers report that JEV (infectious virus and antibody)-positive pigs are found in major JE-endemic areas (China, Japan, Korea, India, Cambodia etc.) even now and GIII-to-GI shift was first observed in Japan, Korea and Vietnam (countries where pig farming is thriving) in 1990s. However, in some areas (Islamic countries) birds may be major reservoir/amplifying hosts of JEV.
5) The reader cannot understand the means of two sentences (1. Prevalenc… 2. Furure threat ...). The left side illustration of mosquito is dispensable. “Shows that there is less chances of JEV transmission from pigs to birds/human/horses by mosquito vectors”
6) L72-73
Several papers pointed out the lower ability of the current JE vaccine against GV JEV. The authors should cite the papers.
7) L113-114
In recent years several reports showed that GI JEV was isolated from cattle and wild boar. The authors cite the articles and also discuss the amplifying host of JEV including these animals.
8) L216-219
No need for detailed data. The Data also described in Table.
9) L317-323, L206-207, L224
No need for detailed data (p-value, the number of animals tested).
10) L325-327
“pCMV” should be deleted.
11) L388
In Malaysia, GV JEV was detected (isolated) only in 1952.
12) L399
“Fig. 1” is correct.
Author Response
Reviewer 1
In this review manuscript the authors focus to the importance of birds as amplifying/reservoir host of JEV and also as key animal in JEV genotype shift from GIII to GI. The review also introduces the historical researches for discovery of birds as the natural amplifying/reservoir host of JEV and for evaluation of sensitivity of birds to JEV infection. The authors also refer to the possibility that birds may be committed mainly to GIII to GI shift in JE-endemic areas.
- First, the reference numbers in the manuscript and Table 1 are inconsistent with those of the references list. The authors should re-check all of the citations in the manuscript.
Ans: Thank you for the correction. We have re-checked all reference numbers in Table 1&2.
2) Only summarized results and discussion of the paper from the authors group (Li C et al #65) are described in paragraphs 9-11 of section 5 (line 335-379). In the paper, the authors did not perform in vivo JEV infection analysis in pigs (only used pig-derived cultured cells). In vitro growth ability of JEV is necessarily reflecting its growth ability in vivo. Fan et al. (#64) showed that GI JEV grew higher than GIII JEV in not only chickens also pigs. Therefore, it is hard to conclude that birds play main role in JEV transmission and genotype shift in most JE-endemic areas.
Ans:
Thank you for the insightful comment. In an initial experiment, Li C et al, infected DEF, swine testicular ST, and a mouse endothelial brain bEnd.3 cells with JEV GI and GIII viruses to analyze the induction of IFN-α and β expression at different time points. There were no significant difference seen in the IFN-α and β production at both the mRNA and protein levels between the GI and GIII strains in either ST or bEnd.3 cells. However, significant differences were observed in DEF cells. The relative production of IFN-α and β in GI-infected DEF were significantly lower than those in GIII-infected DEF cells.
Analysis of the concentrations of IFN-α and β protein in the supernatants revealed that GI strains induced markedly lower levels of IFN-α and β production than GIII strains, thereby counteracting the host antiviral immune response in DEF, but not in ST and bEnd.3 cells, in a species-specific manner. They further confirmed that this difference was species-specific but not cell type-specific by infecting duck embryo kidney cells (DEK), a porcine iliac endothelium cell line (PIEC), and a mouse embryonic fibroblast cell line (MEF) with the GI and GIII strains. Therefore, they carried on further comparison between GI and GIII IFN induction in birds derived cells or duckling and haven’t added pigs in their next level of experiment. We have re-written findings of this study in our manuscript according to your valuable suggestion such as; “Although this experiment is lacking the in vivo data after infection in pigs with these recombinant viruses; however, the above described results explain the host specific differences in the IFN-I induction among GI and GIII strains leads to the replication and host advantages of GI over GIII viruses in birds, that might be a cause of JEV genotype shift”.
3) Table 1 should be reconstructed (made smaller). The column “age” is indispensable? Data of birds infected with each JEV genotype strains should be combined into one row (group). The author should be confirmed the references.
Ans:
Thank you so much for the suggestion.
We have divided Table one into two tables. The studies performed to show the development of infection after JEV infection under lab conditions are presented in table 1. Whereas, the differences in the development of viremia after JEV GI and GIII infection in birds is provided in the table 2.
4) Figure 1 need to be reconstructed. Is there critical evidence that major amplifying host of JEV is now birds not pigs in most JE-endemic areas? In my understand, pigs as well as birds are reservoir and amplifying hosts of JEV and other mammals (cattle, wild boar, bat etc.) also have potential to reservoir/amplifier of JEV. “JEV vaccine mostly used in pigs” is correct? In China and other areas most sentinel pigs are vaccinated? Many papers report that JEV (infectious virus and antibody)-positive pigs are found in major JE-endemic areas (China, Japan, Korea, India, Cambodia etc.) even now and GIII-to-GI shift was first observed in Japan, Korea and Vietnam (countries where pig farming is thriving) in 1990s. However, in some areas (Islamic countries) birds may be major reservoir/amplifying hosts of JEV.
Ans:
Thank for the valuable suggestion. We have reconstructed Fig 1 and corrections have been made according to your suggestion that both Pigs and birds have potential role in the JEV genotype shift.
5) The reader cannot understand the means of two sentences (1. Prevalenc… 2. Furure threat ...). The left side illustration of mosquito is dispensable. “Shows that there is less chances of JEV transmission from pigs to birds/human/horses by mosquito vectors”
Ans: Thank you for the comment. We have modified fig 1 as per suggestion.
6) L72-73
Several papers pointed out the lower ability of the current JE vaccine against GV JEV. The authors should cite the papers.
Ans:
Cited as per suggestion.
Cao L, Fu S, Gao X, Li M, Cui S, Li X, et al. 2016. Low protective efficacy of the current Japanese encephalitis vaccine against the emerging genotype 5 Japanese encephalitis virus. PLoS neglected tropical diseases.10(5):e0004686.
Tajima S, Yagasaki K, Kotaki A, Tomikawa T, Nakayama E, Moi ML, et al. 2015. In vitro growth, pathogenicity and serological characteristics of the Japanese encephalitis virus genotype V Muar strain. Journal of General Virology.96(9):2661-9.
Hegde NR, Gore MM. 2017. Japanese encephalitis vaccines: Immunogenicity, protective efficacy, effectiveness, and impact on the burden of disease. Human vaccines & immunotherapeutics.13(6):1320-37.
7) L113-114
In recent years several reports showed that GI JEV was isolated from cattle and wild boar. The authors cite the articles and also discuss the amplifying host of JEV including these animals.
Ans:
Thank you. We have added information regarding JEV infection in cattle and wild boar in introduction section such as; “Various other species of domestic animals such as cattle, dogs, cats and wild boars can be sub-clinically infected as evidenced by reports of seropositivity to JEV in surveys [1-5]. Previously, a number of seroepidemiological surveys of wild boars antibodies against JEV have reported that a higher percentage of wild boars sera contained JEV antibodies [6]. On Iriomote Island, 44.4% (52 out of 117) of wild boars, in Hiroshima, 68% (17 out of 25) , in Wakayama, 83.3% (30 out of 36), in Ishikawa prefecture, 86.5% (32 out of 37), and in Singapore, 15.16% (10 out of 66) of wild boars serum samples were tested positive for the presence of JEV antibodies [4, 7-10]. Wild boars are closely related to the domestic pigs and does not inoculated with the JEV vaccines; they are thought to play a similar role as a reservoir in the infection cycle of JEV. Recently, JEV GI strains were isolated and characterized from a nonsuppurative encephalomyelitis case of 141-day-old calf and 114 months old cow from Japan, suggesting that JEV GI infection can lead to the development of neurological disorders in cows [11, 12]. These animals do not act as a reservoir/amplification hosts for JEV, however, can play a potential role in future and continuous monitoring is required.”
8) L216-219
No need for detailed data. The Data also described in Table.
Ans:
Thank you. Removed as per suggestion.
9) L317-323, L206-207, L224
No need for detailed data (p-value, the number of animals tested).
Ans:
Thank you. We have removed this information.
10) L325-327
“pCMV” should be deleted.
Ans:
Removed as per suggestion.
11) L388
In Malaysia, GV JEV was detected (isolated) only in 1952.
Ans:
Thank you for the correction. We have changed this sentence such as; Recent detection of more divergent JEV genotype V (8.4% to 10.0% amino acid divergence compared to genotype I-IV) from China [13] and Korea [14] is threatening because……………. .
12) L399
“Fig. 1” is correct.
Ans:
Modified.
References
- Mansfield KL, Hernández-Triana LM, Banyard AC, Fooks AR, Johnson N. 2017. Japanese encephalitis virus infection, diagnosis and control in domestic animals. Veterinary microbiology.201:85-92.
- Mall M, Kumar A, Malik S. 1995. Sero-positivity of domestic animals against Japanese encephalitis in Bareilly area, UP. The Journal of communicable diseases.27(4):242-6.
- Hamano M, Lim CK, Takagi H, Sawabe K, Kuwayama M, Kishi N, et al. 2007. Detection of antibodies to Japanese encephalitis virus in the wild boars in Hiroshima prefecture, Japan. Epidemiology and infection.135(6):974-7. Epub 2007/01/12. doi: 10.1017/S0950268806007710. PubMed PMID: 17217550.
- Komiya T, Toriniwa H, Matsumura T, Takegami T, Nakayama T. 2019. Epidemiological study on Japanese encephalitis virus distribution in Ishikawa prefecture, Japan, by serological investigation using wild boar sera. J Vet Med Sci.81(6):903-5. Epub 2019/04/23. doi: 10.1292/jvms.18-0613. PubMed PMID: 31019149.
- Tan HC, Wang D, Ooi EE, Lee M-A. 2002. Presence of hemagglutination inhibition and neutralization antibodies to Japanese encephalitis virus in wild pigs on an offshore island in Singapore. Acta tropica.81(3):233-6.
- Nidaira M, Taira K, Itokazu K, Kudaka J, Nakamura M, Ohno A, et al. 2007. Survey of the antibody against Japanese encephalitis virus in Ryukyu wild boars (Sus scrofa riukiuanus) in Okinawa, Japan. Japanese journal of infectious diseases.60(5):309.
- Nidaira M, Kyan H, Taira K, Okano S, Oshiro T, Kato T, et al. 2014. Survey of Japanese encephalitis virus in pigs and wild boars on Ishigaki and Iriomote Islands in Okinawa, Japan. Epidemiology and infection.142(4):856-60. Epub 2013/07/09. doi: 10.1017/s0950268813001611. PubMed PMID: 23830350.
- Hamano M, Lim CK, Takagi H, Sawabe K, Kuwayama M, Kishi N, et al. 2007. Detection of antibodies to Japanese encephalitis virus in the wild boars in Hiroshima prefecture, Japan. Epidemiology and infection.135(6):974-7. Epub 2007/01/16. doi: 10.1017/s0950268806007710. PubMed PMID: 17217550; PubMed Central PMCID: PMCPMC2870655.
- Ohno Y, Sato H, Suzuki K, Yokoyama M, Uni S, Shibasaki T, et al. 2009. Detection of antibodies against Japanese encephalitis virus in raccoons, raccoon dogs and wild boars in Japan. J Vet Med Sci.71(8):1035-9. Epub 2009/09/02. doi: 10.1292/jvms.71.1035. PubMed PMID: 19721354.
- Yap G, Lim XF, Chan S, How CB, Humaidi M, Yeo G, et al. 2019. Serological evidence of continued Japanese encephalitis virus transmission in Singapore nearly three decades after end of pig farming. Parasit Vectors.12(1):244-. doi: 10.1186/s13071-019-3501-0. PubMed PMID: 31101069.
- Katayama T, Saito S, Horiuchi S, Maruta T, Kato T, Yanase T, et al. 2013. Nonsuppurative encephalomyelitis in a calf in Japan and isolation of Japanese encephalitis virus genotype 1 from the affected calf. J Clin Microbiol.51(10):3448-53. Epub 2013/07/24. doi: 10.1128/JCM.00737-13. PubMed PMID: 23885004.
- Kako N, Suzuki S, Sugie N, Kato T, Yanase T, Yamakawa M, et al. 2014. Japanese encephalitis in a 114-month-old cow: pathological investigation of the affected cow and genetic characterization of Japanese encephalitis virus isolate. BMC Vet Res.10:63-. doi: 10.1186/1746-6148-10-63. PubMed PMID: 24618225.
- Li MH, Fu SH, Chen WX, Wang HY, Guo YH, Liu QY, et al. 2011. Genotype v Japanese encephalitis virus is emerging. PLoS neglected tropical diseases.5(7):e1231. Epub 2011/07/14. doi: 10.1371/journal.pntd.0001231. PubMed PMID: 21750744; PubMed Central PMCID: PMCPMC3130007.
- Kim H, Cha GW, Jeong YE, Lee WG, Chang KS, Roh JY, et al. 2015. Detection of Japanese encephalitis virus genotype V in Culex orientalis and Culex pipiens (Diptera: Culicidae) in Korea. PloS one.10(2):e0116547. Epub 2015/02/07. doi: 10.1371/journal.pone.0116547. PubMed PMID: 25658839; PubMed Central PMCID: PMCPMC4319795.

Reviewer 2 Report
The manuscript examines the potential role of birds in the genotype shift of Japanese Encephalitis Virus (from G111 to G1) and a literature review is presented discussing the possible role of birds in zoonotic transmission especially in areas where pigs are vaccinated and/or where pig farming is uncommon. Although data is presented to illustrate the emergence of genotype I (GI) as a dominant genotype gradually displacing GIII in Asia, the exact mechanism for this genotype displacement is still unclear. Currently, the conclusions at the end of the paper seem a bit premature as there is little epidemiological or husbandry data presented to support a key role of poultry in the development of the new variant in Asia.
The paper would be improved if it included a diagram, figure or Table outlining the key points raised with regard to the in vitro studies mentioned. This section is quite hard to follow as written.
It would also be good to add more information about pig vaccination, farming practices and the epidemiological data available about outbreaks of JE in the human population. Is there an active surveillance program to study the molecular epidemiology of JE in China ?
Author Response
Reviewer 2
The manuscript examines the potential role of birds in the genotype shift of Japanese Encephalitis Virus (from G111 to G1) and a literature review is presented discussing the possible role of birds in zoonotic transmission especially in areas where pigs are vaccinated and/or where pig farming is uncommon. Although data is presented to illustrate the emergence of genotype I (GI) as a dominant genotype gradually displacing GIII in Asia, the exact mechanism for this genotype displacement is still unclear. Currently, the conclusions at the end of the paper seem a bit premature as there is little epidemiological or husbandry data presented to support a key role of poultry in the development of the new variant in Asia.
Ans:
Thank you so much for the comment. We have modified fig 1 and also amendment have been made in the manuscript to decrease the emphasis on poultry threat. However, we have proposed a hypothesis that further surveillance is required in poultry birds because the emerging JEV GI can develop infection and mortality in future.
- The paper would be improved if it included a diagram, figure or Table outlining the key points raised with regard to the in vitro studies mentioned. This section is quite hard to follow as written.
Ans: Thank you for the comment. We have added Table 3 to provide the summarized information about the mutations in GI and GIII viruses that are might be helping GI viruses to adapt reservoir/amplification host and have effect on JEV transmission.
- It would also be good to add more information about pig vaccination, farming practices and the epidemiological data available about outbreaks of JE in the human population. Is there an active surveillance program to study the molecular epidemiology of JE in China ?
Ans: Thank you for the suggestion. We have added data related to vaccines use in pig, farming practices and recent human cases data in our manuscript such as;
“According to the recent estimated global incidence of Japanese encephalitis, approximately 67,900 JE cases occur per year in the 24 JE endemic countries, for an incidence of 1.8 per 100,000 [1]. Approximately, 33, 900 (50%) of these cases occur in China and 55, 000 (81%) occur in areas with well-established or developing JE vaccination programs, while approximately 12, 900 (19%) occur in areas with minimal or no JE vaccination programs. Approximately 51, 000 (75%) of these cases occur in children aged 0–14 years, which gives an estimated overall annual incidence of 5.4 per 100 000 in this age group [1].
Seroprevalence surveys report that 3.1% to 74% anti-JEV antibodies were present from 1966 to 2016 in swine herds in most of the Asian countries [2]. Pigs can develop high level of viremia that is maintained for two to four days and attract JEV vectors, not only for the virus maintenance/amplification, but also for pig-mosquito-pig or pig-mosquito-human transmission [3-5]. Mostly, pig farms are located close to human dwellings, especially with backyard farming, which is common in Asia; this proximity facilitates the human infection. Four types of JEV vaccines are available in different countries which include a live attenuated vaccine, a live recombinant (chimeric) vaccine, inactivated mouse brain-derived vaccines, and inactivated Vero cell culture-derived vaccines [6-10]. Both JEV inactivated and live-attenuated vaccines are used in pigs to prevent JEV infection in different Asian countries [6, 11, 12].
In addition to pigs as a main JEV reservoir, some studies suggested that other epidemiological systems may exist. A recent autochthonous human JEV cases occurred in Seoul, South Korea, even though no pigs are reared in the city [13]. In 2013, Teng et al. isolated JEV strains from mosquitoes, humans, and pigs, and revealed that the same strain was detected in mosquitoes and human, but not in pigs [14]. Two other studies from Singapore Islands showed that JEV continued to circulate decades after the abolition of pig farming [15, 16]. Thus, there is a need to improve our knowledge of the existence of secondary reservoir host that could explain JEV transmission in areas with no or low pig density.”
Yes, there is an active surveillance program to study the molecular epidemiology of JEV from mosquitoes, pigs and human in China [17-21].
References
- Campbell GL, Hills SL, Fischer M, Jacobson JA, Hoke CH, Hombach JM, et al. 2011. Estimated global incidence of Japanese encephalitis: a systematic review. Bulletin of the World Health Organization.89:766-74.
- Ladreyt H, Durand B, Dussart P, Chevalier V. 2019. How central is the domestic pig in the epidemiological cycle of Japanese Encephalitis Virus? A review of scientific evidence and implications for disease control. Viruses.11(10):949.
- Ricklin ME, Garcìa-Nicolàs O, Brechbühl D, Python S, Zumkehr B, Posthaus H, et al. 2016. Japanese encephalitis virus tropism in experimentally infected pigs. Veterinary research.47(1):1-11.
- Scherer WF, Moyer JT, Izumi T. 1959. Immunologic studies of Japanese Encephalitis Virus in Japan: V. Maternal antibodies, antibody responses and viremia following infection of swine. The Journal of Immunology.83(6):620-6.
- Park SL, Huang Y-JS, Lyons AC, Ayers VB, Hettenbach SM, McVey DS, et al. 2018. North American domestic pigs are susceptible to experimental infection with Japanese encephalitis virus. Scientific reports.8(1):1-8.
- Halstead SB, Jacobson J, Dubischar-Kastner K. 2008. Japanese encephalitis vaccines. Vaccines, 6th ed; Plotkin, SA, Orenstein, WA, Offit, PA, Eds.312-51.
- Organization WH. 2015. Japanese encephalitis vaccines: WHO position paper—February 2015. Weekly Epidemiological Record= Relevé épidémiologique hebdomadaire.90(09):69-88.
- Hills S, Martin R, Marfin A, Fischer M. 2014. Control of Japanese encephalitis in Asia: the time is now. Expert review of anti-infective therapy.12(8):901-4.
- Batchelor P, Petersen K. 2015. Japanese encephalitis: a review of clinical guidelines and vaccine availability in Asia. Tropical diseases, travel medicine and vaccines.1(1):1-8.
- Sasaki O, Karoji Y, Kuroda A, Karaki T, Takenokuma K, Maeda O. 1982. Protection of pigs against mosquito-borne japanese encephalitis virus by immunization with a live attenuated vaccine. Antiviral Research.2(6):355-60. doi: https://doi.org/10.1016/0166-3542(82)90005-5.
- Mansfield KL, Hernández-Triana LM, Banyard AC, Fooks AR, Johnson N. 2017. Japanese encephalitis virus infection, diagnosis and control in domestic animals. Veterinary microbiology.201:85-92.
- Nah J-J, Yang D-K, Kim H-H, Song J-Y. 2015. The present and future of veterinary vaccines for Japanese encephalitis in Korea. Clinical and experimental vaccine research.4(2):130.
- Bae W, Kim JH, Kim J, Lee J, Hwang ES. 2018. Changes of Epidemiological Characteristics of Japanese Encephalitis Viral Infection and Birds as a Potential Viral Transmitter in Korea. Journal of Korean medical science.33(9):e70. Epub 2018/02/15. doi: 10.3346/jkms.2018.33.e70. PubMed PMID: 29441740; PubMed Central PMCID: PMCPMC5811662.
- Teng M, Luo J, Fan J-M, Chen L, Wang X-T, Yao W, et al. 2013. Molecular characterization of Japanese encephalitis viruses circulating in pigs and mosquitoes on pig farms in the Chinese province of Henan. Virus genes.46(1):170-4.
- Ting SHL, Tan HC, Wong WK, Ng ML, Chan SH, Ooi EE. 2004. Seroepidemiology of neutralizing antibodies to Japanese encephalitis virus in Singapore: continued transmission despite abolishment of pig farming? Acta tropica.92(3):187-91.
- Yap G, Lim XF, Chan S, How CB, Humaidi M, Yeo G, et al. 2019. Serological evidence of continued Japanese encephalitis virus transmission in Singapore nearly three decades after end of pig farming. Parasit Vectors.12(1):1-7.
- Deng X, Yan J-Y, He H-Q, Yan R, Sun Y, Tang X-W, et al. 2020. Serological and molecular epidemiology of Japanese Encephalitis in Zhejiang, China, 2015-2018. PLoS neglected tropical diseases.14(8):e0008574-e. doi: 10.1371/journal.pntd.0008574. PubMed PMID: 32853274.
- Pan J-R, Yan J-Y, Zhou J-Y, Tang X-W, He H-Q, Xie R-H, et al. 2016. Sero-Molecular Epidemiology of Japanese Encephalitis in Zhejiang, an Eastern Province of China. PLoS neglected tropical diseases.10(8):e0004936-e. doi: 10.1371/journal.pntd.0004936. PubMed PMID: 27560360.
- Liu W, Fu S, Ma X, Chen X, Wu D, Zhou L, et al. 2020. An outbreak of Japanese encephalitis caused by genotype Ib Japanese encephalitis virus in China, 2018: A laboratory and field investigation. PLoS neglected tropical diseases.14(5):e0008312-e. doi: 10.1371/journal.pntd.0008312. PubMed PMID: 32453787.
- Tao Z, Liu G, Wang M, Wang H, Lin X, Song L, et al. 2014. Molecular epidemiology of Japanese encephalitis virus in mosquitoes during an outbreak in China, 2013. Scientific reports.4:4908-. doi: 10.1038/srep04908. PubMed PMID: 24809635.
- Chai C, Wang Q, Cao S, Zhao Q, Wen Y, Huang X, et al. 2018. Serological and molecular epidemiology of Japanese encephalitis virus infections in swine herds in China, 2006-2012. Journal of veterinary science.19(1):151-5. doi: 10.4142/jvs.2018.19.1.151. PubMed PMID: 28693301.

Round 2
Reviewer 1 Report
1) The authors should re-check the references thoroughly. References #33 and #38 are the same article. In line 353 in the revised manuscript the authors cited two papers (reference #91 and #92). Are the citations correct?
2) lines 98-102, 203-204, 316-318
The numbers of animals (52 out of 117, etc.) should be removed.
3) lines 106-107
“It is unknown that these animals act as a reservoir/amplification hosts for JEV; however, the animals can play a potential role in future and continuous monitoring is required” is better.
4) lines 175-176
Mean of the words “OR” and “nAb” are unclear.
5) lines 214-215
“we have examined the pathogenicity of GI and GIII JEV strains in the Shaoxing ….” Name of each strains is not needed.
6) Tables
In the “level (log10 plaque forming units/mL)” column, “10” (< 101.7, etc.) should be deleted (“< 1.7”)
7) lines 235-241
“… challenged with two GI (SD12 and SH7) and two GIII (SH1 and SH15) JEV strains. In contrast to this, significant mortality was observed in ducklings inoculated with other two GIII JEV strains (N28: 31.7%; SH19: 12.7%)”
8) line 239
“inoculated with a GI strain SD12”
9) Line 263
“bitten by JEV-infected mosquitoes”
10) line 306
“ …by isolating GI viruses from JE patients and field mosquitoes.”
11) lines317-318
“… ducklings, respectively (Table 2).”
“… chicks and ducklings, respectively, had …”
12) lines 319
The authors should unify the expression of titer (10x or log10).
13) lines 321-328
“using recombinant chimeric viruses rGIII/GI UTR, rGIII/GI C-E, …”
14) lines 356
“ between the JEV strains in ducklings (the authors should be add citation).”
15) lines 422
“… GI strains over GIII strains in birds.

Author Response
Reviewer 1
1) The authors should re-check the references thoroughly. References #33 and #38 are the same article. In line 353 in the revised manuscript the authors cited two papers (reference #91 and #92). Are the citations correct?
Ans: Thank you for the correction. We have re-checked all references and removed any of the duplicated citation or irrelevant reference.
2) lines 98-102, 203-204, 316-318
The numbers of animals (52 out of 117, etc.) should be removed.
Ans: Removed as per suggestion.
3) lines 106-107
“It is unknown that these animals act as a reservoir/amplification hosts for JEV; however, the animals can play a potential role in future and continuous monitoring is required” is better.
Ans: Added.
4) lines 175-176
Mean of the words “OR” and “nAb” are unclear.
Ans:
Thank you for the suggestion. We have provided description of OR (An odds ratio that quantifies the strength of the association between two events) and nAb (Neutralizing antibodies)
5) lines 214-215
“we have examined the pathogenicity of GI and GIII JEV strains in the Shaoxing ….” Name of each strains is not needed.
Ans: Removed as persuggestion.
6) Tables
In the “level (log10 plaque forming units/mL)” column, “10” (< 101.7, etc.) should be deleted (“< 1.7”)
Ans: Deleted.
7) lines 235-241
“… challenged with two GI (SD12 and SH7) and two GIII (SH1 and SH15) JEV strains. In contrast to this, significant mortality was observed in ducklings inoculated with other two GIII JEV strains (N28: 31.7%; SH19: 12.7%)”
Ans: Thank you for the correction. We have modified sentence as per suggestion.
8) line 239
“inoculated with a GI strain SD12”
Ans: Added
9) Line 263
“bitten by JEV-infected mosquitoes”
Ans: Modified as per suggestion.
10) line 306
“ …by isolating GI viruses from JE patients and field mosquitoes.”
Ans: Changed
11) lines317-318
“… ducklings, respectively (Table 2).”
“… chicks and ducklings, respectively, had …”
Ans: Changed
12) lines 319
The authors should unify the expression of titer (10x or log10).
Ans: Thank you. We have written titer as 10 x .
13) lines 321-328
“using recombinant chimeric viruses rGIII/GI UTR, rGIII/GI C-E, …”
Ans: Modified
14) lines 356
“ between the JEV strains in ducklings (the authors should be add citation).”
Ans: Added
15) lines 422
“… GI strains over GIII strains in birds.
Ans: Modified
Reviewer 2 Report
The manuscript is much improved with the additional information included. Figure 1 is a bit hard to follow. The use of a picture of a duck may be a bit misleading and not representative of the role of birds in general. Wild aquatic birds are key for the natural circulation of JEV. Poultry species may have a different role as indicated in the manuscript. Table 3 is a good addition but requires a bit more explanation. There are consistent grammatical errors throughout the text with the word 'the' added in many places where it is not required and additions of 'a' or 'an' that are not required.
Author Response
Reviewer 2
Q1. The manuscript is much improved with the additional information included. Figure 1 is a bit hard to follow.The use of a picture of a duck may be a bit misleading and not representative of the role of birds in general. Wild aquatic birds are key for the natural circulation of JEV. Poultry species may have a different role as indicated in the manuscript.
Ans:
Thank you for the suggestion. We have removed duck and added wild birds’ picture in Fig 1.
Q2. Table 3 is a good addition but requires a bit more explanation.
Ans:
Thanks. We have provided more information regarding table 3 such as: JEV genome encodes a single large polyprotein which is consisting of three structural [capsid (C), pre-membrane/membrane (prM), envelope (E)] and seven non-structural proteins (NS1, NS2A, NS2B, NS3, NS4A, NS4B, and NS5) [89, 90]. In this review, we have provided the amino acid variation detail between JEV GI and GIII strains in Table 3 [91]. The sequence alignment data revealed a total of 43 amino acid variations with high conservation rates of 90%–100% and ten amino acid variations with relatively low conservation rates of 50%–89% in their respective genotypes between the GI and GIII strains as demonstrated in Table 3. The amino acid variations were distributed in three structural proteins (C, PrM, and E) and seven non-structural proteins (NS1, NS2A, NS2B, NS3, NS4A, NS4B, and NS5).
Q3. There are consistent grammatical errors throughout the text with the word 'the' added in many places where it is not required and additions of 'a' or 'an' that are not required.
Ans:
Thank you. We have revised the whole manuscript carefully and tried to remove all grammatical/English errors.